# Roles of S-Adenosylmethionine and Its Derivatives in Salt Tolerance of Cotton

**DOI:** 10.3390/ijms24119517

**Published:** 2023-05-30

**Authors:** Li Yang, Xingxing Wang, Fuyong Zhao, Xianliang Zhang, Wei Li, Junsen Huang, Xiaoyu Pei, Xiang Ren, Yangai Liu, Kunlun He, Fei Zhang, Xiongfeng Ma, Daigang Yang

**Affiliations:** 1College of Life Science, Yangtze University, Jingzhou 434025, China; yangli202203@126.com (L.Y.); fyzhao@yangtzeu.edu.cn (F.Z.); 2State Key Laboratory of Cotton Biology, Institute of Cotton Research, Chinese Academy of Agricultural Sciences, Anyang 455000, China; wangxingxing@caas.cn (X.W.); zhangxianliang@caas.cn (X.Z.); liwei@caas.cn (W.L.); huangs19980403@163.com (J.H.); peixiaoyu@caas.cn (X.P.); renxiang@caas.cn (X.R.); liuyangai@caas.cn (Y.L.); hekunlun@caas.cn (K.H.); zhangfei@caas.cn (F.Z.); 3Western Research Institute, Chinese Academy of Agricultural Sciences (CAAS), Changji 831100, China

**Keywords:** salt stress, S-adenosylmethionine, ethylene, polyamines, SAM transporter, cotton

## Abstract

Salinity is a major abiotic stress that restricts cotton growth and affects fiber yield and quality. Although studies on salt tolerance have achieved great progress in cotton since the completion of cotton genome sequencing, knowledge about how cotton copes with salt stress is still scant. S-adenosylmethionine (SAM) plays important roles in many organelles with the help of the SAM transporter, and it is also a synthetic precursor for substances such as ethylene (ET), polyamines (PAs), betaine, and lignin, which often accumulate in plants in response to stresses. This review focused on the biosynthesis and signal transduction pathways of ET and PAs. The current progress of ET and PAs in regulating plant growth and development under salt stress has been summarized. Moreover, we verified the function of a cotton SAM transporter and suggested that it can regulate salt stress response in cotton. At last, an improved regulatory pathway of ET and PAs under salt stress in cotton is proposed for the breeding of salt-tolerant varieties.

## 1. Introduction

Cotton is one of the most important cash crops in the world and the main source of natural fiber. With the continuous growth of the world’s population, cultivated land decreases year by year, and cotton cultivation has gradually shifted to saline land. It is estimated that more than 50% of arable land will be salinized by 2050 [1]. Although cotton is a moderately salt-tolerant crop with a threshold of 7.7 dSm^−1^, long-term and high salt stress affects its growth and yield, especially at the stages of germination and seedling [2,3]. Salt stress reduces the stomatal conductance in cotton, leading to a decrease in photosynthesis and reduced carbohydrate supply in developing boll, which ultimately affects the fiber quality and yield of cotton [4,5]. Currently, although a large number of studies have been conducted to analyze the genetic mechanism of salt tolerance in plants, the specific mechanism of salt tolerance in cotton is still unclear.

As genome sequences have increasingly become available for *Gossypium*, a large number of genes have been identified in response to abiotic stress in cotton [6,7]. For example, the NAC transcription factor gene *GhNAC072* has been overexpressed to improve salt and drought tolerance in Arabidopsis [8], and overexpression of *GhMPK3* alleviated the damage of salt stress in Arabidopsis [9]. S-adenosyl-L-methionine (SAM, also known as AdoMet) is a major methyl donor that not only participates in the methionine (Met) cycle, but also provides aminopropyl for the synthesis of PAs and ET [10,11,12]. A series of studies have revealed that ET is one of the important factors responding to salt stress in many species, as exemplified in Arabidopsis, rice, wheat, cucumber, tomato, and cotton [13,14,15,16,17,18]. PAs, a group of aliphatic amine compounds similar to phytohormones, exist mainly in three endogenous statuses (free, conjugated, bond) and function in the processes of plant development and stress response [19,20]. In soybean, PAs promote hypocotyl elongation by enhancing their own catabolism reactions and increasing the production of hydrogen peroxide (H_2_O_2_) under salt stress [21]. Put2 is a polyamine uptake protein, and overexpression of put2 in tomatoes can improve the antioxidant enzyme activity and salt tolerance of transgenic lines [22]. In addition, PAs respond to the saline–alkali stress in rapeseed through the dynamic adjustment of different PA statuses [23]. However, studies on the regulation of salt stress by PAs are still very few in cotton [24].

SAM, as the precursor of ET and PAs synthesis, needs to be transported to the corresponding organelles through SAM transporters to perform a biological function. SAM transporters were first identified in yeast and humans [25,26]. In Arabidopsis, only two SAM transporters (AtSAMC1 and AtSAMC2) have been identified. The impaired function of *AtSAMC1* will reduce the level of prenyl lipids, which mainly affect the chlorophyll pathway, thereby affecting photosynthesis and reducing tolerance to stress [11]. Thus far, studies on SAM transporters have mainly focused on humans and micro-organisms, with the exception of a few studies on Arabidopsis [27,28,29].

This review focuses on the recent findings on salt tolerance in cotton and systematically introduces the important role of SAM in the response to salt stress through its metabolic derivatives ET and PAs. Furthermore, we identified a putative SAM transporter (*GhSAMC*) from cotton through the transcriptome data of different cotton species and found that it can positively regulate the salt tolerance of cotton. The knowledge presented in this review may be used for breeding elite cotton varieties with high salt tolerance.

## 2. Research Status of Cotton on Salt Stress

Soil salinization affects about 20% of the world’s arable land. This problem continues to exacerbate with global warming, the deterioration of the natural environment, and unreasonable irrigation methods [30,31]. Plants are sessile organisms and cannot escape extreme external environments as animals. Hence, plants have evolved a flexible system that can adjust their morphological, physiological, biochemical, and molecular mechanisms to respond to (a)biotic stresses. Salt stress causes plant cells to suffer from osmotic stress, ion stress, and other secondary stresses, especially oxidative stress [32]. Long-term salt stress will lead to soft and dark leaves in cotton, causing shortened functional periods, early shedding, weak stems, decreased fresh matter and plant height, increased salt concentration, and delayed flowering time [12,33]. In addition, increased relative shedding of flowers and bolls ultimately affects yield and fiber quality in cotton [34] (Figure 1).

### 2.1. Osmotic Regulation

In the early stage of salt stress, plants are subjected to osmotic stress and accumulate a large number of osmomodulating substances, such as inorganic ions (K^+^, Cl^−^, and inorganic salts, etc.), and organic solutes (proline, betaine, soluble sugars, polyols, and polyamines, etc.). Plants try to keep osmotic homeostasis depending on the dynamic adjustment of these osmomodulating substances [35].

### 2.2. Ionic Regulation

With the increase in salt concentration, plant cells accumulate a large amount of Na^+^, causing a K^+^/Na^+^ homeostatic imbalance and Ca^2+^ dysfunction, resulting in ion stress. Plants mitigate ionic toxicity primarily by reducing Na^+^ influx, separating and excreting Na^+^ [36,37]. 

At present, dozens of studies have been carried out on ion transporters in cotton. SOS1, a plasma membrane Na^+^/H^+^ reverse transporter, is responsible for the secretion of Na^+^ in the cytoplasm [35]. Overexpression of the cotton SOS1 gene *GhSOS1* in Arabidopsis revealed that the Na^+^/K^+^ ratio and malondialdehyde (MDA) content decreased in the leaves of transgenic lines treated with salt stress, and the salt-tolerant ability was enhanced [38,39]. NHX, a vacuole Na^+^/H^+^ anti-transporter, is present in roots and leaves and enables excess Na^+^ to be retained in the vacuole. The transport activity of NHX mainly depends on the H^+^ gradient maintained by H^+^-ATPase and H^+^-PPase in the vacuole [40]. In a previous investigation, the overexpression of two cotton homologous genes *GhNHX1* and *GhNHX3D* enhanced the salt tolerance in cotton [41,42]. The K^+^ transporter HAK5 and K^+^ channel AKT1 are responsible for K^+^ uptake in soil, and the Na^+^/K^+^ transporter HKT is responsible for transferring Na^+^ from photosynthetic tissues to roots and extracting Na^+^ from xylem [43,44,45]. While the homologous gene *SbHKT1* was overexpressed in cotton, it can improve the K^+^ absorption ability in transgenic lines and maintain the homeostasis of reactive oxygen species (ROS) [46]. The K^+^ efflux anti-transporter KEA enhances plant salt tolerance by regulating the homeostasis of K^+^ and H^+^ in cells. As an example, it was found that silencing of *GhKEA4* and *GhKEA12* reduced proline and soluble sugar content in cotton leaves and increased sensitivity to salt stress [47]. 

In addition to the above-mentioned genes that have been well studied in regulating Na^+^/K^+^ homeostasis, some new genes in the regulation of the Ca^2+^ and Na^+^ flux have been identified in cotton in recent years, such as the cationic amino acid transporter gene (*GhCAT10D*), cyclic nucleotide-gated channel gene (*GhCNGC1/18/GhCNGC12/31/*), calcium-binding protein gene (*GhCLO6*), annexin gene (*GhANN1/GhANN8b*), and sodium bile acid transporter gene (*GhBASS2*) [48,49,50,51,52]. These observations indicate that cotton has a complex regulatory network to keep ionic homeostasis under salt stress as other plants. Therefore, exploring the key genes in the ionic regulation process is crucial for comprehensively dissecting the regulatory mechanism of salt tolerance in cotton.

### 2.3. Oxidation Regulation

With the extension time of salt stress, osmotic and ionic stresses will induce oxidative stress, causing the accumulation of a large number of ROS in plants. Additionally, chloroplasts, mitochondria, and peroxisomes are the three main sites for ROS accumulation. ROS mainly includes singlet oxygen (^1^O2), superoxide (O_2_^•−^), H_2_O_2_, and hydroxyl radicals (OH•) [53]. Under normal growth conditions, low concentrations of ROS in plants can be used as second messengers, involved in seed germination, growth and development, root growth and gravitropism, programmed cell death, and other processes [54,55]. Under salt stress conditions, plants accumulate a large amount of ROS. High concentrations of ROS can lead to protein denaturation, lipid peroxidation, and nucleic acid damage [56,57]. Although cotton is a relatively salt-tolerant crop, when cotton is in a high-salt environment, a large amount of ROS is produced in the plant. Excessive ROS can affect the growth and development of cotton plants and eventually lead to a decrease in cotton yield and fiber quality [58]. 

In order to avoid the damage caused by ROS, plants synthesize a series of antioxidant enzymes, such as superoxide dismutase (SOD), peroxidase (POD), catalase (CAT), ascorbyl peroxidase (APX), glutathione peroxidase (GPX), and non-enzymatic antioxidants such as carotenoids, phenolic compounds, flavonoids, ascorbic acid (AsA), and glutathione (GSH) [59,60]. In addition, the expression of related genes in plants can regulate the level of ROS and improve the tolerance of plants to salt stress. Recent studies have shown that *ghr-miR414c* can further regulate ROS metabolism by regulating the expression level of *GhFSD1* to affect the salt tolerance of cotton [61]. Cotton WRKY transcription factor *GhWRKY17* can improve the salt sensitivity of transgenic tobacco through ABA signal transduction and regulation of ROS production in plant cells [62]. Under salt stress conditions, the aldehyde dehydrogenase 21 gene (*ScALDH21*) of *S. canadensis* can scavenge excessive ROS in transgenic cotton, thereby improving the salt tolerance of cotton [63]. Silencing the Raf-like MAPKKK gene *GhRaf*19 reduced the accumulation of ROS in cotton, thereby improving the tolerance of plants to salt stress [64].

### 2.4. Signal Transduction Regulation

Over the long evolution, plants have evolved various mechanisms to deal with adversity, such as the accumulation of permeable substances, the regulation of genes associated with Na^+^/K^+^ and Ca^2+^ homeostasis, and the production of antioxidant enzymes and non-enzymatic antioxidants. Each mechanism in above contains a complex signaling pathway, consisting of receptors, secondary messengers, phytohormones, and signal transductors [35]. Of these, the signal transductors include many enzymatic cascades such as mitogen-activated protein kinase (MAPK), calcium-dependent protein kinase (CDPK), G protein, phosphatase, and other components [65,66]. Various transcription factors (TFs), such as MYB [67,68], bZIP [69,70], WRKY [71,72,73], NAC [74,75,76], MYC [77], etc., act as a central regulator and a molecular switch in the signal transduction network under salt stress. These TFs regulate the expression pattern of downstream genes and affect the salt tolerance level by activating or inhibiting their interacting genes [78,79]. In addition, various phytohormones such as abscisic acid (ABA), jasmonic acid (JA), salicylic acid (SA), ethylene (ET), gibberellin (GA), and cytokinin (CK) also participate in the signal transduction of salt stress to regulate the response to salt stress [80,81].

At present, the salt tolerance of cotton can be improved to a certain extent through traditional breeding methods, molecular marker-assisted selection, transgenic breeding, and gene editing. However, salt tolerance in plants is a complex trait controlled by multiple quantitative trait loci (QTL), determined by the response of the whole plant [82]. It is very hard to significantly improve the salt-tolerant ability with changes in a single gene or protein. These require us to further explore the molecular mechanism of cotton salt tolerance to breed varieties with high salt-tolerant capability.

## 3. Role of SAM in Plant Salt Tolerance

As a precursor for the biosynthesis of ET and PAs, SAM is synthesized from the substrates ATP and Met by the catalysis of S-adenosylmethionine synthetase (SAMS) [83]. Related studies have shown that SAMS regulates plant responses to salt stress by promoting the synthesis of SAM. Heterologous expression of potato *SbSAMS* in Arabidopsis led to the accumulation of more SAM, S-adenosyl-L-homocysteine (SAHC), and ET in transgenic lines, showing higher salt and drought tolerance levels [83]. In another study, the overexpression of beet *BvM14-SAMS2* in Arabidopsis improved SAM content and the tolerant ability to salt in transgenic plants, in which the antioxidant system and polyamines’ metabolism played an important role [84].. The advanced analysis found that the biosynthesis of PAs and 1-aminocyclopropane-1-carboxylic acid (ACC) was related to the *CsSAMS1* expression level in transgenic plants [85,86].

## 4. Role of Ethylene (ET) in Plant Growth and Development of Cotton under Salt Stress

When plants are exposed to salt stress, it leads to elevated ROS content in the body, especially including H_2_O_2_ and O_2_^•−^; these increased ROS have a dual effect: when they are at low concentrations, they can act as signaling molecules that mediate salt tolerance, whereas they will produce oxidative damage to plant cells when at high concentrations, thereby disrupting the function of chloroplasts and even triggering plant cell death (PCD) [87,88,89]. As a volatile gas plant hormone, ET is usually considered as another stress hormone in addition to ABA, and its synthesis is induced by various (a)biotic environmental stresses [90,91]. ET also has a dual role in regulating salt stress: one is inhibiting the accumulation of ROS through ET signaling pathways, thereby maintaining the homeostasis of ROS in plants, and the other is promoting ROS production and signal transduction to activate Na^+^/K^+^ transport [92,93,94]. 

### 4.1. ET Biosynthesis

The biosynthesis of ET begins with SAMS catalyzing the conversion of Met to SAM. Next, the SAM is converted into ACC by 1-aminocyclopropane-1-carboxylic acid (ACC) synthetases (ACSs). Finally, using ACC as the substrate, ET is synthesized through ACC oxidases (ACOs) (Figure 2) [95,96,97]. 

ACSs and ACOs are commonly considered key biosynthetic enzymes in ET biosynthesis. Thus far, 12 *ACS* homologous genes have been identified in Arabidopsis [98]. Among these *ACS* genes, *AtACS1* encodes non-functional homodimers or inactivating catalytic enzymes, *AtACS3* is a pseudogene, *AtACS10* and *AtACS12* perform amino transfer functions, and the other 8 ACS genes (*AtACS2*, *AtACS4-9*, *AtACS11*) encode functional ACS [99]. Moreover, the presence of these eight functional active ACS enzymes and their ability to form active heterodimers will increase the diversity of ET biosynthetic reactions and improve the ability to regulate ET production at different developmental and environmental stages [100,101]. A total of 35 GhACS proteins with a conserved C-terminus were identified in upland cotton, whereas the N-terminus of ACS10 and ACS12 were divergent between cotton and Arabidopsis, indicating that they may have a different evolutionary trajectory. QRT-PCR analysis showed that the expression of *GhACS1* was significantly upregulated under salt stress, whereas *GhACS6.3*, *GhACS7.1*, and *GhACS10/12* could respond to cold stress [102]. ACO also plays an important role in ET biosynthesis. Arabidopsis encodes a total of five *ACO* genes, whose expression is regulated by growth and developmental and environmental stresses [103]. Through genome-wide identification, a total of 332 *GhACOs* were identified in upland cotton, and most of them co-expressed with other genes in response to salt and drought stresses. For example, the overexpression of *GhACO106-At* could improve salt tolerance in transgenic Arabidopsis [104]. At present, the ET biosynthetic pathway has been well elucidated in model species, but there is still little research on genes that promote ET biosynthesis and respond to salt stress in cotton, and further exploration is needed in the future.

### 4.2. ET Signaling Transduction

ET can be sensed and bound by ET receptors that are mainly located on the endoplasmic reticulum membrane to stimulate ET response [105,106]. In Arabidopsis, five ET receptors composed of Ethylene response 1 (ETR1), ETR2, Ethylene response sensor1 (ERS1), ERS2, and Ethylene insensitive 4 (EIN4) negatively regulate ET reactions [87,107,108]. ET activates downstream signaling pathways by inactivating these receptors. In the absence of ET, constitutive triple response factor 1 (CTR1) is activated by ethylene receptors and further phosphorylates the C-terminus of Ethylene-Insensitive 2 (EIN2-CEND), causing it to become inactive [109]. Ethylene-Insensitive 3 (EIN3) and EIL1 (EIN3-like1), downstream of EIN2, are degraded by two F-box proteins, EBF1/2 (EIN3 binds to F-box1/2) in the nucleus, thereby blocking the transcription of downstream target genes in the ethylene response [110,111,112]. Under stress conditions, ET increases in plants and is bound to ethylene receptors. As a result, CTR1 is inactivated. Subsequently, EIN2 is dephosphorylated and cleaved to release the EIN2-CEND, which can shuttle into the nucleus with the assistance of EIN2 nuclear-associated protein 1 (ENAP1) and enhance the activity of EIN3/EIL1 [113]. Due to the cascade reaction, downstream target genes that respond to ET, such as ethylene-response factors (ERFs), are regulated and then affect plant growth and development [114,115,116,117]. A summary of ethylene signal transduction pathways is shown in Figure 2.

CTR1 is a Raf-like Ser/Thr protein kinase that negatively regulates the ethylene signal transduction pathway, whereas *GhCTR1* expression was not induced by ET or salt but by SA, GA, and ABA [118]. EIN3/EIL1 are positive regulators of the ethylene signaling pathways and have been characterized in a lot of plants. Under the stimulation of exogenous ACC, the homologous gene *GhEIN3* of EIN3/EIL1 could be induced. A previous study showed that the transgenic Arabidopsis-containing *GhEIN3* had strong salt tolerance, indicating that *GhEIN3* may play a role in plant responses to salt stress by regulating the ROS and ABA pathways [119]. ERFs belong to the APETALA2(AP2)/ERFs family and play an important role in plant stress signaling pathways [120]. Thus far, a series of ERFs-related genes and their functions have been identified and analyzed in cotton, such as the overexpression of *GhERF38* and *GhERF13.12* in Arabidopsis, which improved salt tolerance and revealed dynamic changes in relevant biochemical parameters in transgenic lines [121,122]. In another study, virus-induced silencing of cotton *GhERF12* enhanced the sensitivity of transgenic plants to salt stress and reduced the activity of SOD and POD [123]. Similarly, the silencing of *GhERF4L* and *GhERF54L* in cotton led to reduced resistance to salt stress [124]. In addition, studies had found that each ERF transcription factor contained an AP2 domain, which was the DNA binding domain of ERFs and the key to the regulation of ERFs transcription [125]. A total of 220 genes containing a single AP2 domain sequence has been identified in upland cotton. Transcriptome and qRT-PCR analysis found that *Ghi-ERF2D.6*, *Ghi-ERF-12D.13*, *Ghi-ERF-6D.1*, *Ghi-ERF-7A.6*, and *Ghi-ERF11D.5* could respond to salt stress [126]. Transcriptome meta-analysis and network topology analysis showed that salt stress could increase the expression level of *GhERF109*, and further qRT-PCR analysis showed that *GhERF109* could indeed respond to salt stress [127]. Previous studies have found that *GhERF1*, *GhERF2*, *GhERF3*, *GhERF4*, *GhERF5*, and *GhERF6* mediate plant tolerance to salt and drought [128,129,130,131]. ET correlation factors in cotton are summarized in Table 1.

## 5. Roles of Polyamines (PAs) in Plant Growth and Development under Salt Stress

PAs are a class of low-molecular aliphatic polycations with strong biological activity, widely present in various prokaryotes and eukaryotes [132]. PAs mainly consist of putrescine (Put, a diamine), spermidine (Spd, a triamine), spermine (Spm, a tetramine), and thermospermine (a structural isomer of spermine) [133,134]. In plants, PAs participate in a variety of physiological processes such as organogenesis, embryogenesis, flower development, leaf aging, fruit ripening, etc. [20,135,136]. Under abiotic stress, PAs can not only bind to negatively charged membrane phospholipids to maintain the permeability of cell membranes and the function of membrane proteins but also bind to anionic macromolecules (nucleic acids, proteins, etc.) to affect their physiological functions [137]. Moreover, PAs can partially replace the function of antioxidant enzymes to remove ROS and reduce oxidative damage [138,139]. Up to date, several reports have revealed that PAs regulate plant morphological growth parameters, photosynthetic pigment contents, stress-related indicators, antioxidant enzyme contents, and non-enzymatic antioxidant contents in rapeseed [140], cucumber [141], wheat [142], rice [143] and *Calendula officinalis* L. [144].

### 5.1. PAs Biosynthesis

#### 5.1.1. Put Biosynthesis

Put is a central product in PAs’ biosynthetic pathways [145]. Put mainly has three biosynthetic pathways. The first is the direct synthesis of Put from ornithine (ORN) catalyzed by ornithine decarboxylase (ODC) [146]. The second is to synthesize Put from arginine (Arg) under the ordered catalysis of arginine decarboxylase (ADC), argmatine iminohydrolase (AIH), and N-carbamoylputrescine amidohydrolase (CPA) [147]. In the last, Arg first degraded to citrulline (Cit) catalyzed by Arginine deaminase (ADI) and then catalyzed into Put by citrulline decarboxylase (CDC) decarboxylase (Figure 3) [148].

The Arabidopsis genome contains two ADCs genes (*AtADC1* and *AtADC2*), which have different expression patterns. *AtADC1* was almost not expressed in seeds, roots, and leaves, whereas *AtADC2* was highly expressed in these tissues [149]. Under salt stress, the overexpression of *AtADC2* can regulate the activity of SOD and CAT, thereby improving the salt tolerance of Arabidopsis [150]. The virus-induced silencing of cotton *GhADC2* enhanced plant tolerance to salt stress and reduced H_2_O_2_ content [24].

#### 5.1.2. Spd and Spm Biosynthesis

Put, as a synthetic precursor for Spd and Spm, is catalyzed into Spd by Spd synthetase (SPDS), and then Spm synthase (SPMS) catalyzes Spd into Spm [151,152]. During this process, decarboxyl-SAM (dcSAM), a product catalyzed from SAM by SAM decarboxylase (SAMDC), provides aminopropyl groups for SPDS and SPMS-catalyzing reactions [153] (Figure 3). 

Among these genes encoding key enzymes, SAMDC plays a crucial role in regulating the biosynthesis of Spd and Spm [136]. For example, the overexpression of *FvSAMDC* enhanced salt tolerance in tobacco [154]. In Arabidopsis, the overexpression of *CsSAMDC2* exhibited higher levels of salt and drought tolerance than its wild type [155]. Meanwhile, the ectopic expression of *GhSAMDC3* can improve salt tolerance in Arabidopsis by increasing Spd content and activating salt-tolerance-related genes [156]. In addition, studies have shown that *AhSAMDC* and *TrSAMDC1* can improve plant resistance to salt stress by increasing PAs content and antioxidant enzyme activities [157,158].

### 5.2. PAs Catabolism

The catabolism of PAs in plants relies on the oxidative deamination catalyzed by amino oxidases (AOs), mainly including cupramine oxidases (DAOs) and FAD-dependent polyamine oxidases (PAOs) [152,159]. DAOs convert Put into H_2_O_2_, amino, and Δ1-pyrroline. Δ1-pyrroline can be further converted to γ-aminobutyric acid (GABA) [160]. PAOs mainly decompose Spd and Spm to produce 1,3-diaminopropane and H_2_O_2_ (Figure 3) [161]. Five PAOs (*AtPAO1* to *AtPAO5*) have been identified in Arabidopsis, of which AtPAO1 and AtPAO5 are located in the cytoplasm, and AtPAO2 to AtPAO4 are located in peroxisomes [162,163,164].

PAOs play an important role in plant growth, development, and stress response. For example, *CsPAO2* can interact with *CsPSA3* to improve salt tolerance of cucumber by affecting photosynthesis and promoting PAs conversion [165]. Furthermore, the overexpression of *CsPAO3* in Arabidopsis can promote seed germination and root growth in NaCl-containing media and alleviate growth inhibition induced by salt stress [166]. Liu et al. [167] found that *OsPAO3* in rice was upregulated under salt stress during the germination stage, which increased PAO activity and led to an increase in PAs content of in salt-tolerant strains. Meanwhile, the increased PAs could improve the activity of ROS-scavenging enzymes to eliminate the excessive accumulation of H_2_O_2_ and ultimately elevate the salt tolerance of rice during germination. When *GhPAO3* was overexpressed in Arabidopsis, the salt tolerance was promoted [168]. In addition, PAs can also improve plant resistance to diseases and drought [169,170,171,172]. The above genes related to salt stress in P biosynthesis and catabolism in various crops are summarized in Table 2.

## 6. Cross-Talk between ET and PAs under Salt Stress

ET and PAs share the same precursor SAM, indicating that changes in ET content may affect the homeostasis of PAs [173]. An increasing number of studies have found that salt treatment led to an increase in ET, ACC, Spd, and Spm in peppers, lettuce, spinach, and beetroot, indicating that the SAM pool is high enough to support ET and PAs biosynthesis under conditions, and ET and PAs can coexist without competition [174]. It can be seen that the interaction between ET and PAs depends more on the content of SAM in plants. In addition, co-regulation of endogenous ET and PAs in plants can enhance their tolerances to salt stress. For example, the interaction between *CsCDPK6* and *CsSAMS1* negatively regulates the biosynthesis of plant ET, inhibiting SAMDC, resulting in a decrease in Put conversion to Spd and Spm [86]. Wu et al. [155] overexpressed the PAs related gene *CsSAMDC2* in Arabidopsis to improve salt tolerance, and the expression levels of the ET and ABA responding genes also increased. In a recent report, Freitas et al. [175] found that ET regulated the H_2_O_2_ content derived from PA decomposition enzymes, inducing salt tolerance in maize. Moreover, ET participated in salt-induced oxidative stress in ripening tomato fruits and regulated the metabolic level of PAs [176]. In summary, a series of key genes in the biosynthesis pathways of ET and PAs have been studied to dissect the relationship between ET and PAs. However, the exact interaction mechanism between ET and PAs under salt stress still needs further research.

## 7. Roles of SAM Transporters in Plants

SAM is only synthesized in the cytoplasm and is necessary for the methylation of DNA, RNA, and protein in chloroplasts and mitochondria. Chloroplasts and mitochondria are semi-autonomous organelles and need the help of SAM transporters to transfer SAM into them [177,178]. Two genes, *AtSAMC1* (*At4g39460*) and *AtSAMC2* (*At1g34065*), homologous to yeast (*Sam5p*) and mammalian (*SAMC*) SAM transporters, are present in Arabidopsis. AtSAMC1 is located in the plastid, whereas AtSAMC2 is the dual localization of plastid and mitochondrial membranes [28]. The expression of *AtSAMC2* was almost undetectable in roots, stems, leaves, flowers, and seedlings, whereas *AtSAMC1* was expressed in the above organs and especially had a very high expression in seedlings [179]. Chloroplast SAM transporters are important bonds in many biological processes between cytoplasm and chloroplasts, such as the single-carbon metabolism for maintaining methylation reactions and other SAM-dependent functions in chloroplasts, as well as removing the byproduct SAHC derived from the methylation reaction [180]. Mitochondrial SAM transporters belong to the mitochondrial carrier (MC) family and are important for the methylation reaction in mitochondria [181]. SAM transporters are mainly responsible for the input of SAM in organelles and the output of SAHC to maintain the normal metabolism homeostasis of SAM in plant cells. Once the SAM transporter loses function, it will affect the synthesis of SAM, which is a precursor for ethylene, polyamines, glycine betaine, and lignin. Then, it may ultimately affect the survival of plants in adversity. 

We screened out the salt-tolerant genes through comparative transcriptome analysis in *G. hirsutum*, *G. hirsutum* race *marie-galante*, *G. tomentosum*, and *G. barbadense* (PRJNA933089). Among them, *Gh_A05G2087* (named *GhSAMC*) was found to be a homologous gene of *AtSAMC1* and *AtSAMC2* in cotton after sequence alignment (Appendix A). To investigate whether *GhSAMC* was affected by salt stress in cotton leaves, the TM-1 seedlings of upland cotton were first treated with 250 mM NaCl solution and then sampled at 0 h, 3 h, 6 h, 12 h, and 24 h for qRT-PCR analysis(The primers in the Appendix A). The results showed that *GhSAMC* was affected by salt stress, and its expression revealed a trend that it was rising first, reaching its peak at 6 h, and then falling with the extension of salt stress time (Figure 4E). We further performed virus induced gene silencing (VIGS) to test whether *GhSAMC* was necessary for cotton to tolerate salt stress (Appendix A). When newly emerged leaves were infiltrated with TRV:CLA1, they exhibited an albino phenotype (Figure 4A). The expression levels of *GhSAMC* were measured by qRT-PCR to confirm the silencing efficiency, and the results showed that *GhSAMC* had been silenced completely (Figure 4B). Treated *GhSAMC* silencing (TRV:SAMC) and control (TRV:00) plants with 250 mM NaCl solution at the three-leaf stage, it was found that TRV:SAMC seedlings exhibited more serious symptoms of wilting and lodging than TRV:00 plants (Figure 4C,D). Subsequently, we detected the malondialdehyde (MDA) content and total antioxidant capacity (T-AOC) of VIGS cotton plant leaves (Figure 4F,G), and the results showed that TRV:SAMC accumulated more ROS than TRV:00 seedlings. Therefore, we speculated that *GhSAMC* could regulate the accumulation of ROS under salt stress conditions to improve the salt tolerance of cotton. However, further research is needed to determine whether *GhSAMC* affects the synthesis of ET and PAs by regulating the transport of SAM, thereby regulating salt stress.

Furthermore, we carried out the transcriptomic analysis for leaves of VIGS plants (TRV:SAMC, TRV:00) treated with 250 mM of saline solution (PRJNA937453). Compared with CK (TRV:00), a total of 1169 differentially expressed genes (DEGs) were identified (Appendix A), including 470 upregulated genes and 699 downregulated genes (Figure 5C). GO enrichment analysis revealed that the DEGs were mainly divided into three categories, exemplified as biological processes, cell compositions, and molecular functions (Figure 5A). Moreover, GO categories were concentrated in the functional groups of oxidoreductase activity, oxidative stress response, cell membrane components, and cell walls. These observations indicated that the cell walls and membranes first played a barrier role in response to salt stress, and then the oxidoreductase genes were mobilized into cells to cope with the accumulation of ROS. Thus, the transcriptome data implied that *GhSAMC* may alleviate salt damage by regulating ROS. Plants can respond to salt stress through three ways: osmotic regulation, ion regulation, and oxidative regulation. Therefore, we mapped the differential genes related to these three regulatory methods into a heat map (Appendix A). Among them, there were many DEGs related to oxidation–reduction, mainly including the POD and cytochrome P450 genes. The KEGG results enriched 22 metabolic pathways, including lipid and amino acid metabolism, glycolysis/gluconeogenesis, starch and sucrose metabolism, and carbon metabolism, suggesting that *GhSAMC* may be involved in regulating these metabolic pathways to help cotton cope with salt stress (Figure 5B). 

The tolerance of plants to salt stress is regulated by a series of TFs. Through TFs analysis, 58 TFs related to salt stress were screened, including HD-ZIP, DREB, MYC, Dof, HSF, C2H2, WRKY, MYB, NAC, ERF, and bHLH (Figure 5E). Among them, the TFs families of bHLH, ERF, NAC, and MYB contained significantly more DEGs than other families, including 13, 10, 8, and 8 DEGs, respectively. Previous studies have shown that HD-ZIP [182,183], DREB [184,185], MYC [186,187], Dof [188,189], HSF [190,191], C2H2 [192,193], WRKY [194,195], MYB [196,197], NAC [198,199], and bHLH [200,201] could work alone or synergistically to regulate plant tolerance to salt stress. Additionally, the above TFs could crosslink with ET [202,203,204,205,206,207,208]. For example, the apple NAC transcription factor (*MdNAC047*) enhances plant resistance to salt stress by increasing ET synthesis in plants [209]. *MYB108A* can activate the expression of *ACS1* to promote the formation of ET and ultimately improve the salt tolerance of grapes [210]. The *WRKY29* transcription factor in Arabidopsis positively regulates the expression of the *ACS5*, *ACS6*, *ACS8*, *ACS11*, and *ACO5* genes, thereby promoting ET production [211].

In addition, phytohormones are also involved in the regulation of plant salt stress. Through the analysis of transcriptome data, a total of six phytohormones were screened, including ET, gibberellic acid (GA), auxin, salicylic acid (SA), ABA, and jasmonic acid (JA) (Figure 5D). Among them, ET and GA contained more differential genes, including 10 and 8 DEGs, respectively. The above six phytohormones play an important role in plant growth and development and can synergistically regulate plant tolerance to salt stress [212,213,214]. For example, under salt stress, the interaction between ET and JA inhibits the growth of rice seed roots [215]. Tomato *WRKY23* gene can regulate ET and auxin pathways to improve the tolerance of transgenic Arabidopsis to salt stress [216]. In addition, under normal conditions, GA and auxin can regulate plant growth together with ET, whereas SA, JA, and ABA have antagonistic effects on ET in various stress responses [212].

Finally, combining experimental data and transcriptome data, we summarized the working model for *GhSAMC* in cotton salt stress regulation: *GhSAMC* can regulate the rapid accumulation of SAM transporters when cotton is under salt stress. These SAM transporters accelerate the transport of SAM from cytoplasm to organelle and use SAM as a precursor to further increase the synthesis of ET and PAs. The increase in ET and PAs can inhibit the accumulation of ROS and maintain its homeostasis to alleviate salt stress in cotton (Figure 5F). In this process, related TFs (such as MYB, ERF, WRKY and NAC, etc.) and phytohormones (such as ABA, GA and SA, etc.) can also regulate ET biosynthesis and thus affect plant response to salt stress. The mechanism by which *GhSAMC* regulates plant salt stress needs further exploration.

## 8. Conclusions and Future Perspectives

Salt stress is considered to be one of the main factors limiting crop growth and yield. Therefore, plants have developed various strategies for survival under salt stress, such as osmotic regulation, ion, and ROS homeostasis [217]. Previous studies have shown that oxidative stress caused by salt stress is the most harmful to plants, and our analysis of transcriptome data also shows that a large number of oxidation–reduction-related factors are mobilized when plants are subjected to salt stress. In addition, plant tolerance to salt is regulated by multiple genes, not only a variety of transcription factors (such as EFR, WRKY, MYB, etc.) but also a variety of phytohormones (such as ET, ABA, SA, etc.).

As a general methyl donor for various methylation metabolites, SAM participates in the salt stress response through its derived metabolites (PAs and ET) [218,219]. This review focused on the biosynthesis and signal transduction of ET and PAs in salt stress, as well as the factors involved in these processes. In addition, the cotton SAM transporter gene *GhSAMC*, which is homologous to Arabidopsis *AtSAMC1* and *AtSAMC2*, was also identified. The virus-induced gene silencing experiment suggested that it may have positively regulated the salt tolerance of cotton.

Cotton cultivation has gradually shifted towards saline–alkali land because of the salinization and the need for more arable land to support the ever-increasing population. To improve the tolerance of cotton to saline–alkali stress and increase yield and fiber quality, the following aspects can be considered in the future. First, a certain number of ion transporters have been discovered in cotton, which can alleviate salt stress damage by regulating ion homeostasis. Therefore, it is necessary to further dissect the exact regulatory mechanisms of these ion transporters, and then the knowledge learned from them can be used to improve the salt tolerance of cotton. Second, the impact of ET on cotton growth and development is multifaceted. To date, the regulatory mechanism of ET on cotton fiber development has been well studied, but the specific molecular mechanism of ET in salt stress regulation remains unclear [117,220,221]. Third, it has been proven that PAs can regulate the response to salt stress in many species; however, only a few reports have been found in cotton. Thus, it remains to be studied how PAs function in response to salt stress in cotton. Moreover, as a synthetic precursor for the synthesis of ET and PAs, SAM regulates plant salt stress by controlling the dynamic homeostasis of ET and PAs, thereby enabling SAM biosynthesis to affect responses to salt stress. Our study suggests that silencing *GhSAMC* enhances salt stress damage in cotton. Therefore, in the future, we can use transgenic technology to transfer *GhSAMC* into the cotton or use gene editing technology to knock out the *GhSAMC* gene in cotton, so as to further study the specific molecular mechanism of *GhSAMC* regulating cotton salt stress in cotton. Taken together, SAM and its derivates had very important roles in the response to salt stress, and the investigations summarized in this review revealed the complex biosynthesis and regulatory network of salt tolerance in plants and cotton.

## Figures and Tables

**Figure 1 ijms-24-09517-f001:**
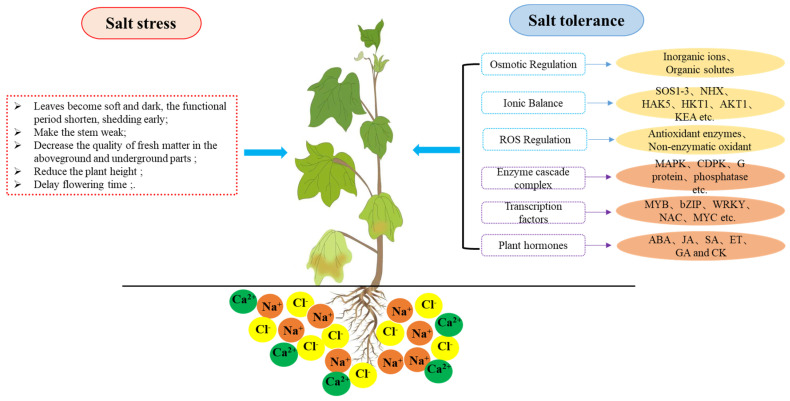
The response to salt stress in cotton. The left part describes the effects of salt stress on the growth and development of cotton; the right part illustrates the related mechanisms and factors of cotton salt tolerance.

**Figure 2 ijms-24-09517-f002:**
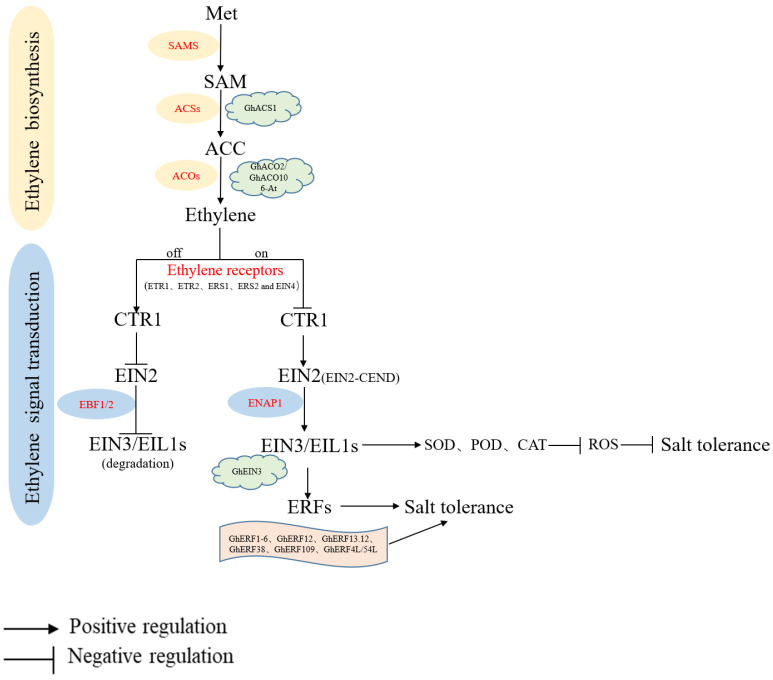
Ethylene biosynthesis and signal transduction. Met: methionine; SAM: S-adenosylmethionine; ACC: 1-Aminocyclopropanecarboxylic Acid; SAMS: SAM synthetase; ACSs: ACC synthases; ACOs: ACC oxidases; CTR1: Constitutive triple response1; EIN2: Ethylene-Insensitive2; EIN3/EIL1: Ethylene-Insensitive3/EIN3-like1; ENAP1: EIN2 nuclear-associated protein1; EBF1/2: EIN3 binding F-box protein1/2; ERFs: Ethylene-response factors.

**Figure 3 ijms-24-09517-f003:**
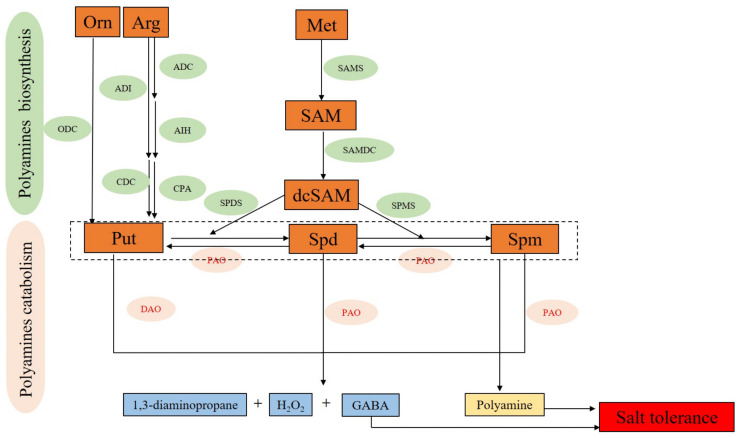
Polyamines biosynthesis and catabolism in plants. Met: methionine; SAM: S-adenosylmethionine; dcSAM: decarboxylated S-adenosylmethionine; GABA: γ-aminobutyric acid; ODC: ornithine decarboxylase; ADC: arginine decarboxylase; CDC: citrulline decarboxylase; AIH: argmatine iminohydrolase; CPA: N-carbamoylputrescine amidohydrolase; ADI: Agmatine deiminase; SAMDC: S-adenosylmethio-Nine decarboxylase; SPDS: spermidine synthase; SPMS: spermine synthase; DAO: Diamine oxidase; PAO: Polymine oxidase.

**Figure 4 ijms-24-09517-f004:**
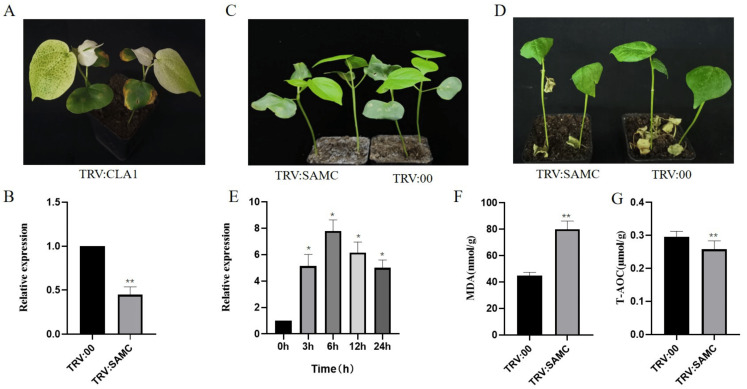
*GhSAMC* silencing in cotton increases the sensitivity to salt stress. (**A**) Albino phenotype of TRV:CLA1 about two weeks after infiltration. (**B**) The silencing efficiency of *GhSAMC*. (**C**) Phenotype of silencing cotton seedlings before salt treatment. (**D**) Phenotype of silencing cotton seedlings after salt treatment. (**E**) Expression of *GhSAMC* over time under treatment with 250 mM saline solution. (**F**) MDA content of silencing cotton seedlings after salt treatment. (**G**) T-AOC content of silencing cotton seedlings after salt treatment (Values are means ± s.e.m (n = 3 biological replicates). Error bars represent the SD of three biological replicates. * *p* < 0.05 (Statistically significant), ** *p* < 0.01 (Statistically highly significant)).

**Figure 5 ijms-24-09517-f005:**
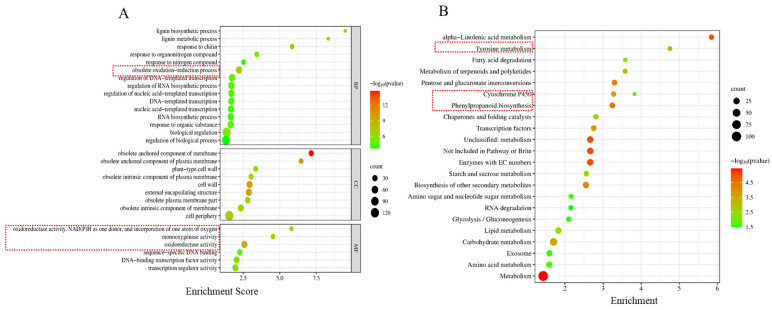
Transcriptome analysis of cotton leaves under salt stress. (**A**) GO enrichment analysis of differentially expressed genes. In the red boxs are some oxidation-reduction related pathways (**B**) KEGG enrichment analysis of differentially expressed genes. The three pathways in the red box, tyrosine metabolism, cytochrome P450, and phenylpropanoid metabolism, play an important role in salt stress. (**C**) The heat map of differentially expressed genes in cotton leaves under salt stress. (**D**) Heatmap of differentially expressed genes related to phytohormone. (**E**) Statistics of the number of differentially expressed transcription factors. (**F**) Model diagram of *GhSAMC* regulation in cotton under salt stress. The red upward arrow indicates an increase in the content of the substance. The red downward arrow indicates an decrease in the content of the substance. ? means that it is uncertain whether the content of the substance is increase or decrease.

**Table 1 ijms-24-09517-t001:** The functions of ethylene-related factors in cotton.

Gene Name	Experimental Methods	Biological Function	Ref.
*GhACS1*	RNA-Seq data analysis and qRT-PCR analysis	Responsed to salt stress	(Li J et al., 2022 [102])
*GhACO106-At*	Overexpression in Arabidopsis	Promoted flowering and increased salt tolerance	(Wei H et al.,2021 [104])
*GhEIN3*	Overexpression in Arabidopsis VIGS in Cotton	Regulated ROS pathway and ABA signaling to response to salt	(Wang X et al., 2019 [119])
*GhERF38*	Overexpression in Arabidopsis	Responsed to salt/drought stress and ABA signaling	(Ma L et al., 2017 [121])
*GhERF13.12*	Overexpression in Arabidopsis VIGS in Cotton	Regulated ROS pathway and ABA signaling to response to salt stress	(Lu L et al., 2021 [122])
*GhERF12*	VIGS in Cotton	Regulated ROS pathway to response to salt stress	(Zhang J et al., 2021 [123])
*GhERF4L/54L*	VIGS in Cotton	Responsed to salt stress	(Long L et al., 2019 [124])
*Ghi-ERF-2D.6/12D.13/6D.1/7A.6/11D.5*	RNA-Seq data analysis and qRT-PCR analysis	Responsed to salt stress	(Zafar M et al., 2022 [126])
*GhERF109*	RNA-Seq data analysis and qRT-PCR analysis	Responsed to salt stress	(Bano N et al., 2022 [127])
*GhERF1*	Semi-qRT-PCR analysis	Responsed to ET, ABA, salt, cold, and drought stress	(Qiao Z et al., 2008 [128])
*GhERF2/3/6*	Semi-qRT-PCR analysis	Responsed to ET, ABA, salt, cold, and drought stress	(Jin L et al., 2010 [129])
*GhERF4*	Semi-qRT-PCR analysis	Responsed to ET, ABA, salt, cold, and drought stress	(Jin L and Liu J., 2008 [130])
*GhERF5*	Semi-qRT-PCR analysis	Responsed to ET, ABA, salt, cold, and drought stress	(Jin L et al., 2009 [131])

**Table 2 ijms-24-09517-t002:** Polyamines (PAs) biosynthesis and catabolism genes in mitigating salt stress in various crops.

PAs	Crops	Genes	Genes Response to Salt Stress	Ref.
Biosynthesis genes	Arabidopsis thaliana	*AtADC2*	Improved SOD and CAT activities	(Fu Y et al., 2017 [150])
	Cotton	*GhADC2*	Increased H_2_O_2_ content and oxidative stress	(Gu Q et al., 2021. [24])
	Fragaria vesca	*FvSAMDC*	Reduced H_2_O_2_ and O_2_^•−^ content	(Kov’acs L et al., 2020. [154])
	Cleistogenes songorica	*CsSAMDC2*	Improved chlorophyll content and Photosynthetic capability	(Wu F et al., 2022. [155])
	Cotton	*GhSAMDC3*	Increased Spd content	(Tang X et al., 2021. [156])
	Peanut	*AhSAMDC*	Improved activities of antioxidant enzymes Increased Spd and Spm content	(Meng D et al., 2021. [157])
	White clover	*TrSAMDC1*	Improved SOD, POD, and CAT activities Reduced MDA and H_2_O_2_ content	(Jia T et al., 2021. [158])
Catabolic genes	Arabidopsis thaliana	*AtPAO1*	Increased ROS and H_2_O_2_ content	(Sagor G et al., 2016. [164])
	Cucumber	*CsPAO2*	Improved activities of antioxidant enzymes Reduced MDA content	(Wu J et al., 2022. [165])
	Cucumber	*CsPAO3*	Improved POD and CAT activities Reduced MDA and H_2_O_2_ content	(Wu J et al., 2022. [166])
	Rice	*OsPAO3*	Increased PAs content Improved Polyamine oxidase activities	(Liu G et al., 2022. [167])
	Cotton	*GhPAO3*	Increased PAs content	(Cheng X et al., 2017. [168])

## Data Availability

The sequencing data for this study have been deposited into the Sequence Read Archive under accession PRJNA933089 and PRJNA937453.

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
