# Peer review of "Roles of S-Adenosylmethionine and Its Derivatives in Salt Tolerance of Cotton"

_ijms, 2023, doi:10.3390/ijms24119517_

Round 1

Reviewer 1 Report

 Review on the manuscript “Roles of S-adenosylmethionine and Its Derivatives in Salt Tolerance of Cotton”.

Salinization is a serious problem for agriculture. The molecular mechanisms of plant resistance to salinity are not fully understood, including in important agricultural crops. In this regards, the topic of this review article is interesting and relevant. The authors hypothesized that the cotton SAM transporter could regulate the response to salt stress in cotton by regulating ROS.  However, this conclusion is questionable. The authors provided very little information on the role of ROS in plants and did not provide information on the role of ROS in cotton (2.3. Oxidation regulation). As evidence that GhSAMC can regulate the accumulation of ROS under salt stress to improve the salt tolerance of cotton, the authors indicate the MDA accumulation (Fig. 4). However, MDA is an indicator of the degree of lipid oxidation and membrane damage as a result of the negative effect of ROS and is considered a marker of oxidative stress rather than plant stress resistance. The authors have presented experimental data (Figures 4 and 5), so they should add the appropriate methods. In Table 2, the Genes Response to Salt Stress should be more specific because the phrase "increased salt tolerance" is not informative. Thus, this manuscript needs major revision.

Author Response

Comments for Reviewer 1

Reviewer 1

1) The authors hypothesized that the cotton SAM transporter could regulate the response to salt stress in cotton by regulating ROS. However, this conclusion is questionable. The authors provided very little information on the role of ROS in plants and did not provide information on the role of ROS in cotton (2.3. Oxidation regulation). As evidence that GhSAMC can regulate the accumulation of ROS under salt stress to improve the salt tolerance of cotton, the authors indicate the MDA accumulation (Fig. 4). However, MDA is an indicator of the degree of lipid oxidation and membrane damage as a result of the negative effect of ROS and is considered a marker of oxidative stress rather than plant stress resistance.

>>>>Response:

We agree with the reviewer’s comment and have added information about the role of ROS in cotton and other plants in page 3. we added reference in page 20: [53-58] , [61-64]. In addition, we concluded that GhSAMC may improve the tolerance of cotton to salt stress by regulating the accumulation of ROS. This conclusion is combined with the accumulation of MDA content and the weakening of total antioxidant capacity (T-AOC) in plants, rather than based on the accumulation of MDA content. Later, we performed Go and KEGG enrichment analysis on transcriptome data (Fig.5), and enriched many processes related to oxidoreductase activity and oxidative stress response, which were also related to ROS accumulation. This also indicates that GhSAMC may be involved in the regulation of ROS accumulation to regulate cotton salt stress.

2) The authors have presented experimental data (Figures 4 and 5), so they should add the appropriate methods.

>>>>Response:

We agree with the reviewer’s comment and have added materials and methods in a new document.

3) In Table 2, the Genes Response to Salt Stress should be more specific because the phrase "increased salt tolerance" is not informative.

>>>>Response:

We agree with the reviewer’s comment and have added more detailed information in Table 2 in page 11 and 12.

Reviewer 2 Report

The manuscript entitled “Roles of S-adenosylmethionine and Its Derivatives in Salt Tolerance of Cotton” by Yang et al. aimed to summarise the role of ET and PAs in regulating plant growth and development under salt stress and verified the function of a cotton SAM transporter and its role in salt stress tolerance in cotton. The authors have tried their best to gather information on the important role of SAM in response to salt stress through its metabolic derivatives ET and PAs. The review has useful information and the authors have done comparative transcriptome analysis to verify their findings.

However, I have the following major comments for the authors:

1.      Please rephrase some of the words in the left part that describes the effects of salt stress on the growth and development of cotton in Figure 1. For example- weak the stems…

2.      In Figure 2, the authors have mentioned the importance of EIN2, EIN3 and a number of ERFs in cotton in response to salt stress. Since you have done comparative transcriptomic analysis, have you found any upregulation of any of those TFs and other TFs? If so, you can explain your results in response to salt stress in cottons. Is there any crosstalk with other TFs?

3.      Please provide the differential gene expression data in a supplementary file.

4.   Have you found any crosstalk with other phytohormones in your comparative transcriptomics data?

5.      Since you have transcriptomics data, some part of the review on other plants can be nicely verified based on your own experimental data.

6.      Figure 5 can be improved in more detail based on your transcriptomic data.

Minor revision required. 

Author Response

Reviewer 2

1) 1. Please rephrase some of the words in the left part that describes the effects of salt

stress on the growth and development of cotton in Figure1For example- weak the stems...
>>>>Response:

We agree with the reviewer’s comment and have improved the description in Figure 1 in page 5.

2) In Figure 2,the authors have mentioned the importance of EIN2 EIN3 and a number of

ERFs in cotton in response to salt stress. Since you have done comparative transcriptomic analysis, have you found any upregulation of any of those TFs and other TFs? If so you can explain your results in response to salt stress in cottons. Is there any crosstalk with other TFs?

>>>>Response:

In comparative transcriptomics data, we have found some up-regulated and down-regulated ERFs. There are some other cross-linked TFs, including HD-ZIP, DREB, MYC, Dof, HSF, C2H2, WRKY, MYB, NAC and bHLH. We counted the TFs related to salt stress in Figure 5E in page 16 and made relevant descriptions in page 14.

3) Please provide the differential gene expression data in a supplementary file.

>>>>Response:

We agree with the reviewer’s comment and have provided the differential gene expression data in supplementary Table S2.

4) Have you found any crosstalk with other phytohormones in your comparative transcriptomics data?

>>>>Response:

In comparative transcriptomics data, We found that ethylene, gibberellin, auxin, salicylic acid, abscisic acid and jasmonic acid were cross-linked with each other. We mapped differentially expressed genes related to phytohormones into a heat map in Figure 5D in page 16. We have also made relevant descriptions in page 14.

5) Since you have transcriptomics data some part of the review on other plants can be nicely verified based on your own experimental data

>>>>Response:

We agree with the reviewer’s comment and have improved the description in page 14 and page 16.

6)  Figure 5 can be improved in more detail based on your transcriptomic data.

>>>>Response:

We agree with the reviewer’s comment and have improved the description about phytohormones and TFs in Figure 5D and 5E in page 16. In addition, we also counted differentially expressed genes related to osmotic regulation, ion regulation and oxidative regulation and drew a heat map in Supplementary Figure S2 (due to the picture is relatively large, it is placed in the attachment).

Round 2

Reviewer 1 Report

The authors took into account the comments and recommendations. There are minor comments to the manuscript.

Fig 1 is double and fig. 5 partially double

 Materials and methods - The authors should add information about illumination (PAR,  μmol/(m2 s)  ) and nutrient solution for plant growth.

The authors should add statistical analysis for determination *P<0.05,**P<0.01 in Fig. 4

Author Response

Comments for reviewer 1

Reviewer 1

  • The authors have mentioned in line 453-457 that in comparative transcriptomics data, we have found some up-regulated and down-regulated ERFs. There are some other cross-linked TFs including HD-ZIP, DREB, MYC, Dof, HSF, C2H2, WRKY, MYB, NAC and bHLH.

However, the authors have just mentioned the results, but there is no discussion on the role of these TFs in response to salt stress. Please briefly explain the role of these TFs and the crosstalk between them and whether there is any link with ethylene/ERF TFs.

>>>>Response:

We agree with the reviewer’s comment and have added information about the role of these TFs and the crosstalk between them and their link with ethylene in page 13(in line 456-465). we added reference in page 23-25: [181-210]. The relevant description words as follows:

Previous studies have shown that HD-ZIP[181,182], DREB[183,184], MYC[185,186], Dof[187,188], HSF[189,190], C2H2[191,192], WRKY[193,194], MYB[195,196], NAC[197,198] and bHLH[199,200] can work alone or synergistically to regulate plant tolerance to salt stress. And the above TFs can crosslink with ET[201-207]. For example, apple NAC transcription factor (MdNAC047) enhances plant resistance to salt stress by increasing ET synthesis in plants[208]. MYB108A can activate the expression of ACS1 to promote the formation of ET, and ultimately improve the salt tolerance of grapes[209]. The WRKY29 transcription factor in Arabidopsis positively regulates the expression of ACS5, ACS6, ACS8, ACS11 and ACO5 genes, thereby promoting ET production[210].

  • Regarding the phytohormones, the authors have just mentioned the results. You need to briefly explain their roles and whether any crosslink with ethylene.

>>>>Response:

We agree with the reviewer’s comment and have added information about the role of phytohormones and their link with ethylene in page 13(in line 470-476). we added reference in page 23-25: [211-215]. The relevant description words as follows:

The above six phytohormones play an important role in plant growth and development, and can synergistically regulate plant tolerance to salt stress[211-213]. Such as, under salt stress, the interaction between ET and JA inhibits the growth of rice seed roots[214]. Tomato WRKY23 gene can regulate ET and auxin pathways to improve the tolerance of transgenic Arabidopsis to salt stress[215]. In addition, under normal conditions, GA and auxin can regulate plant growth together with ET, while SA , JA and ABA have antagonistic effects on ET in various stress responses[211].

3) Figure 5F can be improved based on your revised TFs and phytohormone data.

>>>>Response:

We agree with the reviewer’s comment and have improved Figure 5F in page 15.

(According to the results of transcriptome data analysis, we focused on selecting differentially expressed genes with significant differences and representatives to draw a heat map of transcription factors and phytohormones in Fig5F.)

Reviewer 2 Report

Thank you authors for revising the manuscript.

However, I have the following major comments for the authors to improve the quality of their manuscript.

1.     The authors have mentioned in line 453-457 that ‘In comparative transcriptomics data, we have found some up-regulated and down-regulated ERFs. There are some other cross-linked TFs, including HD-ZIP, DREB, MYC, Dof, HSF, C2H2, WRKY, MYB, NAC and bHLH. …’

However, the authors have just mentioned the results, but there is no discussion on the role of these TFs in response to salt stress. Please briefly explain the role of these TFs and the crosstalk between them and whether there is any link with ethylene/ERF TFs.

2.     Regarding the phytohormones, the authors have just mentioned the results. You need to briefly explain their roles and whether any crosslink with ethylene.

3.     Figure 5F cab be improved based on your revised TFs and phytohormone data.

The language is okay.

Author Response

Reviewer 2

  • Fig 1 is double and fig. 5 partially double

>>>>Response:

We agree with the reviewer’s comment and have deleted the repeated pictures in Fig. 1 and Fig. 5.

  • Materials and methods - The authors should add information about illumination (PAR,umol/(m's) ) and nutrient solution for plant growth.

>>>>Response:

We agree with the reviewer’s comment and have added information about illumination (PAR,umol/(m's) ) and nutrient solution for plant growth in Materials and methods. The relevant description words as follows:

  1. hirsutum Texas Marker-1 (TM-1) was cultivated by soil culture method with hoagland nutrient solution by coolaber, grow in the greenhouse with 14 h light /10 h dark , 50% relative humidity and 200 µmol/(m2 s) illumination intensity at 25â—¦C.

3) The authors should add statistical analysis for determination *P<0.05**P<0.01 in Fig. 4

>>>>Response:

We agree with the reviewer’s comment and have added information about statistical analysis for determination *P<0.05**P<0.01 in Fig. 4 in page 12(in line 430-431). The relevant description words as follows:

*P<0.05(Statistically significant), **P<0.01(Statistically highly significant)).

Round 3

Reviewer 2 Report

I do not have any more comments.

It is okay.